# Safety of Adjuvanted Recombinant Herpes Zoster Virus Vaccination in Fragile Populations: An Observational Real-Life Study

**DOI:** 10.3390/vaccines12090990

**Published:** 2024-08-29

**Authors:** Maria Costantino, Valentina Giudice, Giuseppina Moccia, Walter Longanella, Simona Caruccio, Giuliana Tremiterra, Pio Sinopoli, David Benvenuto, Bianca Serio, Francesca Malatesta, Nadia Pecoraro, Emilia Anna Vozzella, Riccardo Rossiello, Giovanni Genovese, Francesco De Caro

**Affiliations:** 1Department of Medicine, Surgery, and Dentistry, University of Salerno, 84081 Baronissi, Italy; vgiudice@unisa.it (V.G.); gmoccia@unisa.it (G.M.); scaruccio@unisa.it (S.C.); psinopoli@unisa.it (P.S.); dbenvenuto@unisa.it (D.B.); fmalatesta@unisa.it (F.M.); npecoraro@unisa.it (N.P.); fdecaro@unisa.it (F.D.C.); 2University Hospital “San Giovanni di Dio e Ruggi d’Aragona”, 84121 Salerno, Italy; dmp.ruggi@sangiovannieruggi.it (W.L.); bianca.serio@sangiovannieruggi.it (B.S.); direzione.sanitaria@sangiovannieruggi.it (E.A.V.); giovanni.genovese@sangiovannieruggi.it (G.G.); 3D.E.A. Nocera/Pagani/Scafati Hospital, Local Health Authority, 84016 Salerno, Italy; giuliana.tremiterra@aslsalerno.it; 4Epidemiology and Prevention Unit, Local Health Authority, 84014 Salerno, Italy

**Keywords:** AEFI, herpes zoster, vaccine hesitancy, vaccination

## Abstract

Background: Vaccination is the most effective strategy for preventing infectious diseases and related complications, and proving its efficacy is crucial for its success and adherence, especially for newly introduced vaccines, such as adjuvanted recombinant herpes zoster virus vaccination (RZV). In this observational real-life study, we recorded adverse effects following immunization (AEFIs) after RZV administration in frail populations. Methods: A total of 271 subjects underwent RZV at Vaccination Center, University Hospital “San Giovanni di Dio e Ruggi d’Aragona”, Salerno, Italy. Most subjects were solid organ transplant recipients (kidney, 77.1%; liver, 4.8%). Demographics, clinical data, and AEFIs (type, duration, and medications used) were recorded. Results: Overall, 37% of participants reported at least one AEFI following the first dose, predominantly pain at the injection site (60%), while 41% did so after the second dose (pain at the injection site in 62% of cases). Medications were more frequently used for AEFI treatment after the second dose (28%) rather than after the first dose (13%) (*p* = 0.01). After stratification by sex, females experienced AEFIs more frequently than males, particularly local skin reactions. Conclusions: Our study added evidence of safety and tolerability of the adjuvanted recombinant RZV in frail adults.

## 1. Introduction

Vaccination is one of the most effective strategies for preventing infectious diseases and their complications, as vaccines avoid 2–3 million deaths worldwide each year. Moreover, a higher global vaccination coverage could further reduce infectious-disease-related deaths by an additional 1.5 million/year [1,2]. Increasing the success of vaccination programs is mandatory to monitor efficacy and safety of available vaccines, particularly of newly introduced interventions. A recombinant herpes zoster (HZ) virus vaccine (RZV), an adjuvanted glycoprotein-based vaccine, has been recently included in the Italian National Vaccination Prevention Plan and is a significant advancement in HZ reactivation prevention in fragile individuals, such as those with multiple comorbidities and reduced fitness [3,4,5]. Even though RZV is associated with adverse events incidence—such as local reactions at the injection site and systemic symptoms like fever and fatigue—these events are generally mild to moderate in severity and typically resolve within a few days [6]. This indicates that RZV has a favorable safety profile with no significant increase in serious adverse events, consistent with findings for other vaccines [7].

RZV has demonstrated high efficacy in preventing herpes zoster and its complications, particularly in elderly and immunocompromised populations. This high efficacy is largely attributed to its potent adjuvant, AS01B, that enhances immune responses. Clinical trials have consistently reported vaccine efficacy rates of over 90%, even in older adults, that are significantly higher than that of other HZ vaccines. Indeed, RZV has shown an efficacy of 97.2% in adults aged 50 years or older and 91.3% in those aged 70 years or older [8], and its efficacy remains above 90% in adults over 70 years old [9]. Furthermore, RZV has shown a high efficacy of 87.2%, even in immunocompromised individuals, such as those with hematological malignancies, where other vaccines are generally less effective [10].

In frail populations, HZ reactivation can progress to a severe disease and can have lasting pain, known as post-herpetic neuralgia, vision or hearing dysfunctions, and/or peripheral neuropathy. RZV can effectively reduce the risk of HZ reactivation and is recommended for all immunocompetent adults aged ≥65, individuals aged ≥50 with immunodeficiency (e.g., HIV-positive, people undergoing dialysis, or autoimmune diseases), and patients aged ≥18 with severe immunodeficiency, such as during malignant hematological neoplasms, hematopoietic stem cell or solid organ transplant recipients, or under JAK inhibitors [3,11].

Although efficacy and safety of RZV have been established in clinical trials, post-marketing surveillance is crucial to increase vaccine coverage and reduce hesitancy [5]. Vaccine hesitancy is ~15% worldwide [1,2,12]. In 2019, the World Health Organization (WHO) identifies vaccine hesitancy as one of the top 10 threats to global health, contributing to an increase in infectious disease cases worldwide [1,13]. This hesitancy can stem from various factors, such as misinformation about vaccine risks and benefits and beliefs that vaccines are unsafe or may cause other diseases, such as autism [14,15,16]. Additionally, healthy individuals often avoid vaccination because they believe their risk of contracting severe disease is very low, and accessibility issues can further impede vaccination, particularly for working individuals who need to take a day off to visit the vaccination center. Furthermore, frail individuals are often under-represented in clinical trials; therefore, real-life evidence is important to confirm of efficacy and safety of vaccinations in the general population.

In this study, we provided a surveillance investigation on RZV at a single Italian center, reporting types of AEFIs, their severity, duration, and frequency, also stratified by gender. We primarily aim at contributing to enhance evidence on safety of RZV in a real-life setting, thus increasing confidence in vaccination and its coverage.

## 2. Materials and Methods

### 2.1. Recombinant Vaccine (RZV)

Recombinant zoster vaccine (RZV; GSK, London, UK) is a recombinant, adjuvanted vaccine designed to prevent HZ. This vaccine is composed of a glycoprotein E antigen derived from the varicella-zoster virus, combined with an AS01 adjuvant system to enhance immune responses [11], and is effective in reducing incidence and complications of HZ in adults over 50 years old, as well as in individuals with immunodeficiency starting from the age of 18. This vaccine is administered as a two-dose series, with the second dose given 2 to 6 months after the first dose. Its safety profile has been investigated in several extensive clinical trials, and is generally well tolerated. Common side effects include injection site reactions, muscle pain, and fever [11]. In Italy, RZV is provided at no cost by the government to all elderly citizens, reflecting the commitment to improving public health by reducing HZ incidence among the senior population.

### 2.2. Population and Study Design

A total of 271 frail Caucasian adults (mean age, 56 ± 13 years; range, 20–84 years; M/F, 66%/34%) were included in this observational study who were recommended by a healthcare specialist to receive RZV and were referred to the Vaccination Center, University Hospital “San Giovanni di Dio e Ruggi d’Aragona”, Salerno, Italy, between December 2022 and March 2024. Two doses of RZV were administered intramuscularly, with an interval of 2–6 months between doses or even after 1 month in severe immunocompromised patients. This study was conducted in accordance with the Declaration of Helsinki and protocols approved by our local Ethics Committee “Campania Sud”, Naples, Italy (n. 185_r.p.s.o./2022). A specific Case Report Form (CRF) was used for patient data collection. Inclusion criteria were immunocompetent subjects aged ≥65 years; immunodeficient patients aged ≥50, including HIV-positive individuals, those under dialysis, with type 2 diabetes, or systemic vasculitis; subjects aged ≥18 with severe immunodeficiency, such as patients with hematological malignances or receiving hematopoietic stem cells or solid organ (kidney or liver) transplant; and signed informed consent. Exclusion criteria were immunocompetent subjects aged <65 years and subjects who did not consent to sign informed consent. Clinical characteristics are summarized in Table 1. Most enrolled subjects had received solid organ transplantation, especially kidney (77.1%) or liver (4.8%) or hematopoietic stem cell transplantation (3.7%) (Table 1). Patients receiving solid organ transplants, particularly kidney and liver transplants, were on various immunosuppressive therapies. Kidney transplant recipients commonly used medications such as prednisone, cyclosporine, tacrolimus, mycophenolate mofetil, everolimus, and sirolimus. Liver transplant recipients were treated with similar immunosuppressants, including tacrolimus, cyclosporine, mycophenolate mofetil, and prednisone. Patients who underwent hematopoietic stem cell transplantation were often on immunosuppressants such as cyclosporine, mycophenolate mofetil, steroids, and anti-cytomegalovirus prophylaxis. Dialysis patients were typically managed with antihypertensive medications and erythropoietin, while HIV-positive individuals often received antiretroviral therapy. The study considered these pharmacological treatments in the context of vaccine administration. For kidney and liver transplant recipients, as well as hematopoietic stem cell transplant patients, particular attention was given to the timing of vaccine doses in relation to immunosuppressive therapy to optimize the immune response. Dialysis and HIV-positive patients were also carefully managed to ensure appropriate scheduling and monitoring, addressing any potential interactions to enhance vaccine efficacy.

### 2.3. Demographic and AEFI Data Collection

A questionnaire was administered to subjects receiving RZV vaccine at the time of the second vaccine dose administration to collect the occurrence of adverse events after the first vaccine dose administration and by telephone contact after 2 months from the second dose. Italian and European guidelines were employed to classify adverse events as AEFIs or severe AEFIs (SAEFIs) [17,18]. Clinical data (underlying disease, comorbidity, vaccine indication, injection site, batch number, and expiration date of administered vaccine) and AEFIs or SAEFIs (type, duration, and medications used as treatment) were recorded. A patient was considered lost to follow-up if the second dose was not administered within six months from first administration. However, AEFIs or SAEFIs were collected by telephone after two months from the first dose administration.

### 2.4. Statistical Analysis

Demographic characteristics of the study population were assessed by descriptive analysis. Continuous variables were presented as mean ± standard deviation (SD), and two-group comparisons were performed using an unpaired or paired *t*-test for normally distributed data. Categorical variables were analyzed using χ^2^ test. A *p* value < 0.05 was considered statistically significant. Data collection and analysis were carried out using SPSS 23.0 statistics package.

## 3. Results

### 3.1. RZV Vaccine Is Safe with Local Mild AEFIs

After the first RZV dose (n = 271), 37% of patients (n = 101) reported at least one AEFI (Table 2), particularly those aged between 51 and 60 years (35.6%). The most frequently reported AEFI was pain at the injection site (n = 61; 60%), followed by fever (n = 28; 28%), local skin reactions such as rash, swelling, and itching (n = 25; 25%), and fatigue–exhaustion (n = 20; 20%). At the time of the second dose administration, 13 patients declined to complete the vaccination cycle due to vaccine reluctance, 1 patient passed away, another died due to disease-related complications, 1 kidney transplant recipient experienced worsening renal conditions, and 1 patient did not receive the second dose due to the appearance of a severe systemic skin rash and hives, which had also been observed with other medications. This case of severe rash was an outlier and does not reflect the overall safety profile.

Among 271 subjects who received the first dose, 254 of them (93.7%) completed the vaccination cycle, reporting AEFIs in 41% of cases (n = 105). No significant differences were observed between AEFI occurrence after the first and second dose (41% vs. 37%; *p* = 0.34), and pain at the injection site was the most common symptom (n = 65; 62%), followed by fever (n = 41; 39%), fatigue–exhaustion (n = 20; 19%), and local skin reactions (n = 13; 12%) (Table 2). Medications were more frequently used for AEFI treatment after the second dose (n = 29, 28% vs. n = 13, 13%, second vs. first dose; *p* = 0.01), and paracetamol, alone or in combination with probiotics, was the most used drug (Figure 1). Although the use of medications was associated with better management of AEFIs, leading to a significantly shorter duration of symptoms, expressed in hours (h), after the second dose compared to the first dose (mean duration ± SD, 53 ± 76 h vs. 90 ± 157 h, respectively; *p* = 0.03) (Table 2), the data do not conclusively indicate that the increased medication use directly caused this reduction in duration. Other factors may also contribute to this outcome, such as individual variability in immune response, the booster effect after the first dose, patient experience in managing symptoms, and psychological factors. Similarly, local reactions after the first dose lasted longer compared to systemic manifestations (mean duration ± SD, 110 ± 176 h vs. 51 ± 57 h, respectively; *p* = 0.03) (Table 2).

### 3.2. Recurrence of AEFIs after First Dose of RZV Is Frequent While Not Threatening

Among those subjects with an AEFI after the first dose (n = 101), 95 of them (94.1%) completed the vaccination cycle, while 5 patients were refused the second dose and 1 had a systemic severe skin rash and hives. Among 95 fully vaccinated subjects, AEFI recurrence was reported in 62 cases (65.3%), with similar characteristics of those described after the first dose, except for local skin reactions, less frequently observed after the second dose (*p* = 0.01) (Table 3). Next, we compared patients with AEFI recurrence (n = 62) with those who experienced AEFIs only after the second dose (n = 43), no significant differences were observed in types, incidences, and duration (Table 3). However, the duration of local AEFIs was longer than that of systemic AEFIs, as reported for reactions observed after the first administration.

### 3.3. Females More Frequently Experience Local Skin AEFIs

Finally, to investigate sex differences in AEFI incidence, type, and duration, subjects were stratified by sex and results were compared, showing that females more frequently experienced AEFIs than males after both the first (47% vs. 32%; *p* = 0.01) and second dose (51% vs. 36%; *p* = 0.03) (Table 4). However, no significant differences were observed between sexes when divided by types of AEFIs, except for local skin reactions at the site of injection, which were more frequent in females after both the first (36% vs. 16%; *p* = 0.02) and second dose (24% vs. 3%; *p* = 0.01). Similarly, the mean duration was similar between both sexes.

## 4. Discussion

Vaccination is one of the most important achievements of modern medicine, as it has significantly contributed to global public health by preventing infectious-disease-related deaths. In this real-life observational study, we added evidence of the safety of RZV, recently included in the Italian PNPV 2023–2025, also in elderly and frail populations with comorbidities [8,19]. Indeed, AEFI occurrence rate was similar to that reported in clinical trials, including immunocompetent and immunocompromised subjects, and in other real-life study conducted on chronic disease patients [8,20,21,22].

AEFI severity has been classified according to the World Health Organization (WHO) criteria [23]. Mild events are defined as those not requiring specific treatments and resolved spontaneously, such as mild pain at the injection site or mild systemic symptoms like fever or malaise. Moderate AEFIs require symptomatic treatments without significantly interfering with daily activities, such as moderate pain, fever, or skin reactions. Severe events need urgent medical intervention or hospitalization.

In our study, all adverse events were transient and generally categorized as mild to moderate in severity. Although we observed a single case of severe systemic skin rash in one patient, this was a single report and not representative of the overall safety profile of this vaccine, as the majority of reported adverse events were mild or moderate, with no other severe AEFIs. Pain or local skin reactions at the injection site were the most commonly reported symptoms, followed by fever and fatigue, especially in subjects aged 51–60 years, with rates similar to those previously described in literature [8,24,25,26,27]. Indeed, studies on safety and reactogenicity of RZV consistently indicate that AEFIs, such as injection site reactions, pain, and fatigue, are typically transient and of mild to moderate intensity [11,27]. In addition, in our study, we found that females reported AEFIs more frequently than males after both the first and second dose, with more intense local reactions at the injection site and longer duration.

In summary, our results strongly support the administration of RZV due to its safety profile also in frail populations. No severe adverse events, including Guillain–Barré syndrome (GBS)—a rare but serious neurological condition considered one of the most significant adverse events potentially associated with RZV—were observed, in accordance with safety profiles reported in other studies [28]. Although the association between GBS and RZV is a serious concern, it remains a rare event in the context of overall safety of this vaccine.

Antipyretic/analgesics are the most used medications worldwide for treatment of fever and pain after vaccine inoculation, and Centers for Disease Control and Prevention recommends ibuprofen or paracetamol to ease discomfort [29,30], without affecting efficacy, vaccine failure, low antibody production, and antibody blunting [29]. Conversely, a negative effect could be observed when these drugs are given as prophylaxis or with novel antigen vaccination not on boosting administration [29,31,32], as thermal stress induced by antigen stimulation could increase immune responses following vaccination [29,33]. Therefore, antipyretic/analgesics might lower vaccine response, while no conclusive results are reported in humans [34], especially in immunocompromised populations [29]. In our real-life study, we showed a significant increase in medication use, especially paracetamol, to control adverse events after the second dose, likely shortening their duration, in particular local AEFIs. Overall, RZV was safe after first and second dose administration, and adverse events observed after the second dose had similar severity and characteristics of those reported after the first dose. Our results highly supported RZV administration because of its safety profile also in frail populations. Indeed, Italian Regulations have included only RZV vaccine, while not the live attenuate formulation, because of its contraindication in immunocompromised subjects despite lower AEFI rates [35]. Of note, we showed adverse event rates of adjuvanted RZV in our frail patients similar to those reported in the immunocompetent Australian general population [35].

Our results added evidence to RZV safety and could also be useful in reducing vaccine hesitancy [19,36,37,38], a growing global problem with a worldwide incidence of 15%, similar to that reported in our study. Specifically, 12.6% of participants declined the RZV vaccine when initially offered and 6% refused to complete the vaccination series after the second dose. This led to incomplete vaccination and potentially reduced efficacy. This reluctance to receive RZV was attributed to various factors, including perceptions of vaccines as unnecessary, a general reluctance to receive additional vaccines, distrust stemming from previous allergies to other medications, or a general mistrust of vaccines.

In low-income countries, nonadherence to vaccination is related to low education level, lack of insurance coverage, economic hardship, or poor accessibility [39,40,41], while, in high-income countries, vaccination hesitation is mainly due to misconceptions, news reports linking autism and autoimmune diseases to vaccination, and lack of trust in medicine, scientific research, and the authorities [41,42,43]. In our real-world study, we confirmed that, in high-income countries, such as Italy, regardless of patients’ education level, the main reason for vaccine hesitancy is lack of confidence in vaccination, often due to suspected or documented AEFIs to previous vaccines or after the first RZV dose. In a phase 3, two-arm, randomized, double-blind, placebo-controlled, multicenter trial conducted on solid tumor patients, AEFI recurrence rate was similar between treatment (22.5%) and placebo (21.0%) groups, and severe reactions were very rare (<1%), while local skin reactions at the injection site were frequent [20]. Similarly, in a retrospective population-based 3-month follow-up study from the South of Italy, a comparable geographic area to that in which our study was conducted, AEFIs have been reported in 42.7/100 cases in a group of patients with chronic onco-hematological, cardiovascular, endocrine-metabolic, or rheumatological diseases, with local reactions being the most observed (35.6/100 cases), especially in females, as also reported in our study [21].

Our study has some limitations: (i) the observational nature of designed investigation and inclusion of a relatively small sample size; (ii) a short follow-up for identification of long-term safety outcomes; and (iii) vaccine hesitancy in this study was under-reported because we missed those patients who received RZV recommendation while they never referred to our Vaccination Center.

## 5. Conclusions

In conclusion, our study demonstrates that RZV vaccine is both safe and well tolerated in frail adults, including those with complex health conditions, such as transplant recipients, individuals undergoing dialysis, and those with hematological malignancies. Our findings support the use of RZV in this high-risk population, showing that adverse events are generally mild to moderate, aligning with existing literature on RZV safety profile. Furthermore, our data underscore that vaccine hesitancy remains a significant issue, with 12.6% of participants declining the first dose and 6% refusing to complete the vaccination series. This highlights the need for targeted strategies to address vaccination concerns. Despite these encouraging results, further research with larger sample sizes and extended follow-up periods is needed to confirm these findings and assess the long-term impact of RZV vaccination on this vulnerable population. Moreover, continued efforts to educate and engage patients about benefits of vaccination are crucial to overcome vaccine hesitancy and improve vaccination rates.

## Figures and Tables

**Figure 1 vaccines-12-00990-f001:**
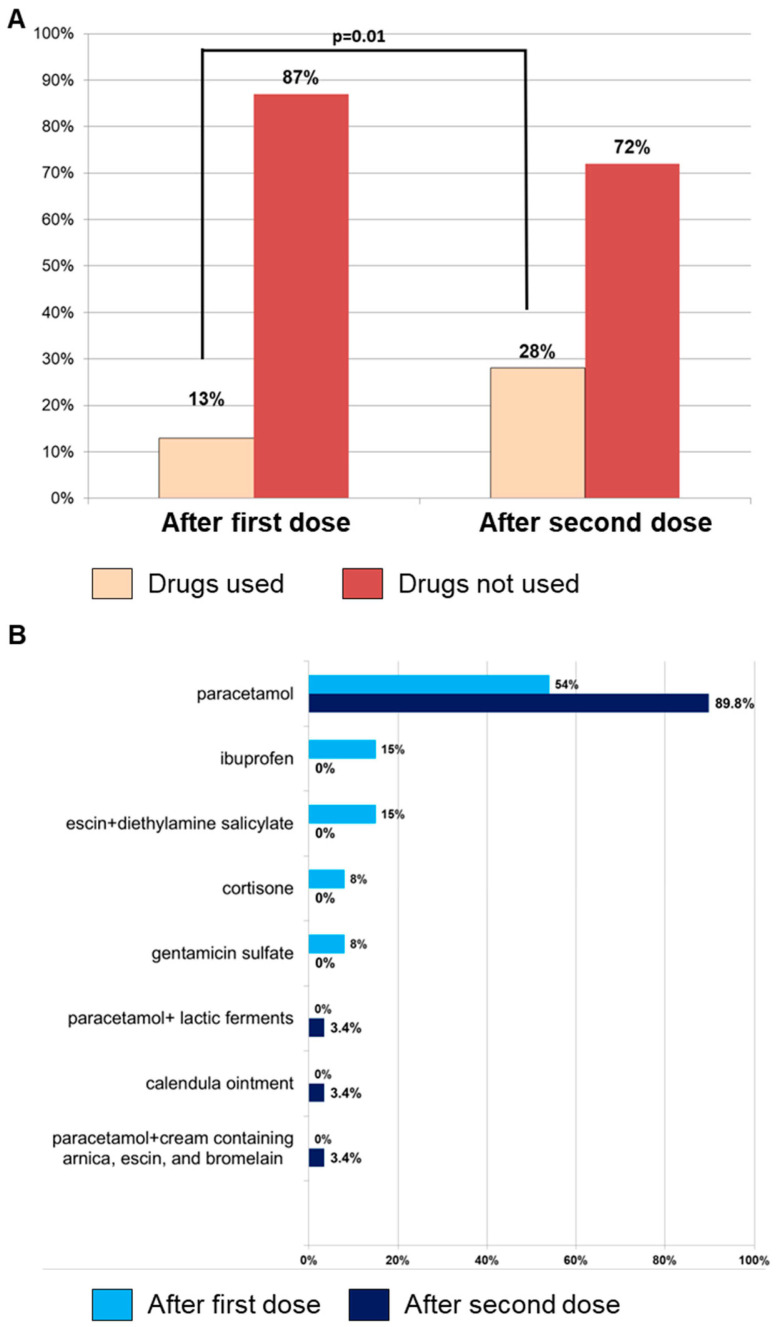
Adverse events following immunization (AEFI) incidence and medication usage. (**A**) Incidence of AEFIs occurring after the first (n = 101) and second (n = 105) RZV dose and, in each group, the number of subjects who used medications for their treatment is reported. (**B**) Types of drugs used for treating AEFIs after the first or second dose.

**Table 1 vaccines-12-00990-t001:** Characteristics of patients at first dose (n = 271) or with completed RZV (n = 254).

	First RZV Dose (n = 271)	Completed RZV (n = 254)
Mean age ± SD, years (range)	56 ± 13 (20–84)	56 ± 12 (30–84)
Males/Females, n (%)	178 (66)/93 (34)	165 (65)/89 (35)
Fragility, n (%)		
Type 2 diabetes	2 (0.7)	2 (0.8)
Hematological malignancies	4 (1.5)	4 (1.6)
Hematopoietic stem cell transplant	10 (3.7)	9 (3.5)
Dialysis	19 (7.0)	16 (6.3)
Kidney transplant	209 (77.1)	196 (77.2)
Liver transplant	13 (4.8)	13 (5.1)
HIV infection	11 (4.1)	11 (4.3)
Systemic vasculitis	1 (0.4)	1 (0.4)
Immunocompetent subjects aged ≥65 years	2 (0.7)	2 (0.8)

**Table 2 vaccines-12-00990-t002:** Incidence and type of AEFI after first dose or at completed RZV.

	First RZV Dose (n = 271)	Completed RZV (n = 254)	*p* Value
Incidence, n (incidence rates)	101 (37)	105 (41)	0.34
Age ranges, years, n (%)			
20–30	3 (3.0)	6 (6)	
31–40	8 (7.9)	11 (10)	
41–50	18 (17.8)	22 (21)	
51–60	36 (35.6)	30 (29)	
61–70	27 (26.7)	22 (21)	
≥71	9 (9.0)	14 (13)	
Total AEFI duration (hours), mean ± SD,	90 ± 157	53 ± 76	0.03
Local AEFI duration (hours), mean ± SD, (n)	110 ± 176 (77)	65 ± 88 (70)	0.06
p value within group	0.03	0.05	
Systemic AEFI duration (hours), mean ± SD (n)	51 ± 57 (46)	41 ± 33 (58)	0.27
Clinical manifestations, n (%)			
Pain at the inoculation site	61 (60)	65 (62)	0.82
Fever	28 (28)	41 (39)	0.09
Fatigue—Exhaustion	20 (20)	20 (19)	0.89
Local skin reactions	25 (25)	13 (12)	0.02
Headache	6 (6)	8 (8)	0.63
Bone pain	3 (3)	9 (9)	0.09
Gastrointestinal symptoms	4 (4)	9 (9)	0.17
Paresthesia	1 (1)	0 (0)	0.31
Stomatitis	1 (1)	0 (0)	0.31
Rash at nose–lips and forehead	1(1)	0 (0)	0.31
Cutaneous systemic rash	1 (1)	0 (0)	0.31
Flushes	0 (0)	1 (1)	0.33
Muscle aches	0 (0)	2 (2)	0.16

**Table 3 vaccines-12-00990-t003:** Type and incidence of AEFIs in patients with AEFI recurrence (n = 62) and in those who have experienced AEFIs only after the second dose (n = 43).

AEFIs Clinical Manifestations, n (%)	After First Dose (n = 62)	After Second Dose (n = 62)	*p* Value	After Only Second Dose (n = 43)	*p* Value
Pain at the inoculation site	36 (58)	37 (60)	0.86	30 (70)	0.29
Fever	21 (34)	22 (35)	0.85	19 (44)	0.37
Fatigue—Exhaustion	12 (19)	15 (24)	0.51	5 (12)	0.11
Local skin reactions	23 (37)	10 (16)	0.01	4 (9)	0.31
Headache	6 (10)	6 (10)	0.99	2 (5)	0.34
Bone pain	3 (5)	6 (10)	0.30	2 (5)	0.34
Gastrointestinal symptoms	2 (3)	3 (5)	0.65	6(14)	0.10
Paresthesia	1 (2)	0 (0)	0.32	-	-
Muscle aches	0 (0)	1 (2)	0.32	1 (2)	0.79
Flushes	0 (0)	0 (0)	-	1 (2)	0.23
Total AEFI duration (hours), mean ± SD (n)	76 ± 125 (78)	58 ± 87 (74)	0.31	40 ± 33 (54)	0.15
Local AEFI duration (hours), mean ± SD (n)	97 ± 153 (45)	72 ± 113 (40)	0.62	51 ± 33.4 (31)	0.32
*p* value within group	0.09	0.15	-	0.20	-
Systemic AEFI duration (hours), mean ± SD (n)	48 ± 62 (33)	42 ± 33 (34)	0.40	39 ± 34 (23)	0.74

**Table 4 vaccines-12-00990-t004:** Type and incidence of AEFIs after first dose and completed RZV stratified by sex.

	First RZV Dose			Completed RZV		
	Females n = 44	Males n = 57	*p* Value	Females n = 45	Males n = 60	*p* Value
AEFIs, (%)	47%	32%	0.01	51%	36%	0.03
Clinical manifestations n (%)						
Pain at the inoculation site	24 (55)	38 (67)	0.22	30 (67)	37 (62)	0.59
Fever	8 (18)	20 (35)	0.06	16 (36)	25 (42)	0.53
Fatigue—Exhaustion	11 (25)	9 (16)	0.25	11 (24)	9 (15)	0.22
Local skin reactions	16 (36)	9 (16)	0.02	11 (24)	2 (3)	0.01
Headache	4 (9)	2 (4)	0.24	4 (9)	4 (7)	0.67
Bone pain	2 (5)	1 (2)	0.41	4 (9)	4 (7)	0.67
Gastrointestinal symptoms	1 (2)	3 (5)	0.45	3 (7)	6 (10)	0.55
Stomatitis	1 (2)	0 (0)	0.25	0 (0)	0 (0)	-
Cutaneous systemic rash	1 (2)	0 (0)	0.25	0 (0)	0 (0)	-
Paresthesia	0 (0)	1 (2)	0.38	0 (0)	0 (0)	-
Muscle aches	0 (0)	0 (0)	-	2 (4)	0 (0)	0.10
Flushes	0 (0)	0 (0)	-	0 (0)	1 (2)	0.38
Rash at nose–lips and forehead	0 (0)	1 (2)	0.38	0 (0)	0 (0)	-
Total AEFI duration (hours), mean ± SD (n)	82 ± 114 (54)	93 ± 167 (69)	0.68	56 ± 35 (57)	51 ± 86 (71)	0.68
Local AEFI duration (hours), mean ± SD, (n)	92 ± 130 (34)	120 ± 202 (45)	0.48	64 ± 46 (34)	60 ± 114 (37)	0.85
*p* value within group	0.40	0.07		0.07	0.39	
Systemic AEFI duration (hours), mean ± SD (n)	65 ± 80 (20)	42 ± 26 (24)	0.19	44 ± 33 (23)	42 ± 36 (34)	0.83

## Data Availability

Data are contained within this article.

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
