# Peer review of "Safety of Adjuvanted Recombinant Herpes Zoster Virus Vaccination in Fragile Populations: An Observational Real-Life Study"

_vaccines, 2024, doi:10.3390/vaccines12090990_

Round 1

Reviewer 1 Report

Comments and Suggestions for Authors

This clinical study is a report about the safety of the Shingrix vaccine when administered to a fragile population of Italian patients. This study is important because the authors do not list any involvement of the GSK company in study and the authors themselves do not list any conflicts. Because Shingrix is proving to be an extremely effective vaccine, these clinical studies from different countries around the world are critical for validation of GSK-supported studies. See 3 comments for improvement. 

1.Methods, lines 63-82. Please add another section to Methods. Call this section the recombinant vaccine. Include a brief English summation of the reference 7 in Italian. Please add a sentence to state whether Shingrix is given at no cost by the Italian government to all elderly citizens of Italy. 

2.Discussion section, lines 164-175. Guillain-Barre syndrome (GBS). Mention that GBS is perhaps the most serious adverse event following Shingrix injection. Please state that no cases of GBS were seen in this small Italian study. Please add this reference about GBS and Shingrix and discuss in the text. Reference: R. Goud et al, JAMA Internal Medicine 181:1623, 2021. 

3.Conflict of interest statement, line 253. Because of its importance, please state specifically that none of the authors have been paid by the GSK company for any prior services or speaking engagements. 

.           

Reviewer 2 Report

Comments and Suggestions for Authors

In the article “Safety of Adjuvanted Recombinant Herpes Zoster Virus Vaccination in fragile populations: an observational real-life study“ the authors describe and discuss in detail adverse events after vaccination with Shingrix in immunocompromised patients. The data is presented in a well-structured and understandable manner.

Some general amendments:

Shingrix is a highly efficient vaccine due to a potent adjuvant. The efficacy could be mentioned explicitly in the introduction, e.g., and discussed shortly in relation to the number of adverse events and in correlation to adverse events after use of other vaccines.

Some minor amendments:

Results

Page 3, line 123-128: The duration of AEFI is longer after the second vaccination and after the second vaccination more medication was used. However, does the data provide the conclusion that the use of more medication is the reason for shorter duration of symptoms? Please clarify.

Page 3, line 114: One patient with “severe systemic skin rash…” vs. page 6, line 170: “all manifestations… mild or moderate severity”. Please describe more precise.

Discussion

Page 6, line 181: First appearance of acetaminophen. In the results paracetamol is used. Please use only one name.

References

Page 9, citation 1 to 3: Please cite original publications for the facts mentioned.

Reviewer 3 Report

Comments and Suggestions for Authors

The article refers to the report of unwanted effects and the use of anti-inflammatory therapy after herpes virus vaccination. The rationale is adequate, and the methodology, in general, is sufficient. However, some issues can enhance the manuscript and should be included: 1) most patients are kidney transplant patients, which pharmacologically differ from others in the same group. Therefore, I suggest to include a table with the current treatment used. 2) The vaccine and the frequent side effects in the normal population are not referred to as comparison; 3) vaccine hesitance is discussed but not proven. The authors should clarify: 4) They are a high-risk population; therefore, other vaccines are expected to be used; which ones? 5) the conclusion should be modified since it is not informative. 
